# Effects of Chewing Gum on Satiety, Appetite Regulation, Energy Intake, and Weight Loss: A Systematic Review

**DOI:** 10.3390/nu17030435

**Published:** 2025-01-25

**Authors:** Claudia Jiménez-ten Hoevel, Elisabet Llauradó, Rosa M. Valls, Maria Besora-Moreno, Judit Queral, Rosa Solà, Anna Pedret

**Affiliations:** 1Functional Nutrition, Oxidation, and Cardiovascular Diseases Group (NFOC-Salut), Facultat de Medicina i Ciències de la Salut, Universitat Rovira i Virgili, 43201 Reus, Spain; claudia.jimenez@urv.cat (C.J.-t.H.); rosamaria.valls@urv.cat (R.M.V.); mariadelaserra.besora@urv.cat (M.B.-M.); judit.queral@urv.cat (J.Q.); anna.pedret@urv.cat (A.P.); 2Institut Investigació Sanitària Pere i Virgili (IISPV), 43204 Reus-Tarragona, Spain; 3Hospital Universitari Sant Joan de Reus, 43204 Reus, Spain

**Keywords:** chewing gum, appetite regulation, satiety, energy intake, weight loss, obesity

## Abstract

Background: New approaches for the management of obesity, a worldwide problem and a major determinant of disability and mortality, are needed. Mastication influences appetite and satiety mechanisms via actual food or sham feeding. However, the effect of mastication of chewing gum, a type of sham feeding, on appetite regulation has not yet been elucidated. Objectives: Our aim was to evaluate the influence of chewing gum on appetite regulation, satiety, energy intake, and weight loss via randomized controlled Trials. Methods: This study was conducted in accordance with the 2020 PRISMA guidelines, and the protocol was registered in PROSPERO (CRD42023432699). Electronic databases MEDLINE^®^/PubMed, Scopus, and Cochrane Central Register of Controlled Trials were searched from July 2023 to September 2024. The quality of each included study was assessed using the Cochrane risk of bias tool, RoB 2. Results: A total of eight articles with nine RCTs were included in this systematic review. Seven out of nine RCTs evaluated appetite regulation. Five out of seven RCTs reported a significant suppressing effect of hunger, three out of five RCTs reported a significant reduction in desire to eat, and three out of four reported a significant reduction in the desire to eat a sweet snack, all of them compared to the control group. However, the effects on satiety, energy intake, and weight loss are not conclusive. Conclusions: Chewing gum could be a promising non-pharmacological tool for obesity management through appetite regulation; however, further research, with sustained RCTs evaluating the sustained effects of gum chewing on appetite and weight management, is needed.

## 1. Introduction

Obesity is a worldwide problem and a leading cause of cardiovascular diseases and diabetes [1]. Since 1990, worldwide adult obesity has doubled. In 2022, 43% of adults aged 18 years and over lived with overweight and 16% lived with obesity [2].

Overweight and obesity are defined as abnormal or excessive fat accumulation that may impair health [2]. The characteristics of our current society facilitate obesity by overconsumption, such as the increase in ultraprocessed food consumption, the limited availability of healthy sustainable food at affordable prices, and a lack of safe and easy mobility into daily life [2].

Although people with obesity and without obesity do not differ in the frequency of eating, people with obesity consume a greater number of calories than people without obesity [3,4]. This suggests that satiety and appetite regulation may influence meal size, being an important contributor to energy overconsumption and obesity. Moreover, the orosensory stimulation provided by food contributes to the appetite and compensatory dietary responses. The hardness of food, mastication, and time of chewing contribute to satiation and satiety signals [5]. Thus, a greater bite size and increased speed of eating decrease satiation and are considered risk factors for obesity [6,7]. Satiation is described as the process leading to meal termination; it controls meal size and is influenced by several feedback mechanisms such as declining food preference and gastric fullness [8]. However, satiety is defined as the inhibition of further eating as well as the suppression of the feeling of hunger and occurs as a consequence of having eaten [8]. Satiety is influenced by a number of pre- and post-absorptive feedback mechanisms such as macronutrient composition, energy density, physical structure and the sensory qualities of food, and the gastrointestinal peptides that are released following food consumption (CCK, GLP-1, and ghrelin) [4,8].

Additionally, appetite is considered as the sensation that motivates intake and can be present even in the absence of a physiological need [4]. Hunger and the desire to eat represent approach behaviors indicative of appetite or readiness to eat [9].

In recent years, the importance of chewing has been implicated as a strategy for preventing overeating, since reduced masticatory function is associated with obesity [10,11], and a positive correlation has been established between increased speed of eating, higher body mass index (BMI), and higher energy intake [12,13]. Chewing stimulation could reduce subjective appetite and influence the metabolic appetite regulation system [14,15] and, consequently, prevent weight gain. Correspondingly, previous systematic reviews and meta-analyses concluded that increasing masticatory activity reduces food intake and subjective appetite and increases satiety [16,17,18].

Chewing could be stimulated by actual feeding or by sham feeding. Sham feeding is described as to view, smell, taste, and chew food without ingesting it, promoting gastrointestinal peristalsis and an increase in the salivatory rate [19]. Chewing gum is considered as a type of sham feeding. Actual feeding was compared to sham feeding, and similar results were obtained showing that hunger and preoccupation with food were significantly reduced and fullness significantly increased after feeding or sham feeding [20]. This suggests that chewing stimulation itself, with or without food ingestion, might reduce appetite. Mattes et al. concluded that gum mastication increased resting energy expenditure when compared to no gum chewing, suggesting that chewing without ingestion of food would have an impact on total energy intake and energy expenditure [5].

Chewing gum is very widespread and accepted amongst all populations [21]. In a survey conducted in 2021 in a Spanish population, 14.2% consumed sugars and sweets daily, including chewing gum [22]. Chewing sugar-free gum would increase satiety and reduce subjective appetite without adding extra energy. Furthermore, it could help suppress cravings for high-energy snacks or meals and be considered a promising tool for managing obesity.

Previous systematic reviews evaluated the effects of actual food and sham feeding on satiety, appetite regulation, and energy intake, but not exclusively with chewing gum [16,17,18]. To the best of our current knowledge, no systematic review has explicitly assessed the influence of chewing gum on satiety, appetite regulation, and energy intake. Furthermore, there has not been an update on the subject, and the existing systematic reviews date back more than five years.

Thus, it was hypothesized that chewing gum affects appetite regulation. Accordingly, the present study aimed to evaluate the effectiveness of chewing gum on satiety, appetite regulation, energy intake, and, consequently, weight loss determined via randomized control trials (RCTs) through a systematic review.

## 2. Materials and Methods

This systematic review included RCTs that evaluated the effect of chewing gum on satiety, appetite regulation, energy intake, and/or weight loss in adults. This study is reported in accordance with the 2020 Preferred Reporting Items for Systematic Reviews and Meta-Analyses (PRISMA) guidelines [23], and the protocol was registered in PROSPERO International Prospective Register of Systematic Reviews (CRD42023432699).

### 2.1. Literature Research

The following electronic databases were searched for eligible studies: MEDLINE^®^/PubMed, Scopus, and Cochrane Central Register of Controlled Trials. The keywords used were chewing gum, mastication, satiety, satiety regulation, appetite, appetite regulation, hunger, body weight, BMI, weight loss, waist circumference (WC), obesity, energy intake, and caloric intake. The search strategy is described in Appendix A. 

### 2.2. Eligibility Criteria and Study Selection

The inclusion criteria were as follows: (a) RCTs; (b) limited to the English language and human studies; (c) published from 2000 to 2024; (d) adults aged ≥ 18 years from the worldwide general population, healthy or with some comorbidities; (e) assessed only one, two, or all of the outcomes; and (f) mastication of a low-calorie chewing gum, for all timings, frequencies, and doses.

In contrast, the exclusion criteria were as follows: (a) children and adolescents (<18 years); (b) older adults (>65 years); (c) interventions that did not meet one or more of our inclusion criteria; (d) exposure to chewing but not chewing gum; (e) mastication of other low-calorie food; (f) studies that only included individuals with a specific health condition; (g) studies that reported no relevant data on the mentioned outcomes; (h) systematic reviews and/or meta-analyses; (i) case–control studies; (j) cohort studies; (k) protocols; (l) grey literature; (m) correspondence letters; (n) government statistics summaries; (o) book chapters; (p) dissertations; and (q) conference summaries.

The Population, Intervention, Comparison, Outcomes and Study (PICOS) criteria (Table 1) were used to define the inclusion and exclusion criteria of the RCT studies, and the search was conducted from July 2023 to September 2024.

### 2.3. Data Extraction

We selected the included studies using Covidence systematic review software, Veritas Health Innovation, Melbourne, Australia (available at www.covidence.org, accessed on 14 July 2023).

Published studies were selected in the first stage based on a title and abstract screening according to the inclusion and exclusion criteria. In the second stage, full-text articles that had passed the first stage of screening were assessed. Finally, in the third stage, only RCTs that met all the criteria described previously were included for data extraction and quality assessment.

All titles and abstracts were screened for inclusion by two independent researchers (C.J.-t.H. and A.P.). After this first screening, the full text of the studies was read and assessed according to the predefined inclusion criteria by one independent researchers (C.J.-t.H.). In case of discrepancy, a second author helped to extract the article’s data (A.P.). Discrepancies were resolved after discussion with a third researcher (E.L.). When any necessary information for inclusion was missing from any study, we contacted the authors to request it.

The variables collected from the included studies were (a) authors; (b) year of publication; (c) country; (d) study design; (e) sample size; (f) age of participants; (g) sex of participants; (h) health state of participants (with or without obesity); (i) aim of the intervention; (j) type of intervention and setting; (k) duration of the intervention and characteristics (time of mastication, frequency a day, etc.); (l) tools used for assessing main and additional outcomes; (m) changes in outcomes.

### 2.4. Outcomes

The main outcomes included were (1) subjective ratings of hunger and desire to eat as markers of appetite regulation, (2) subjective ratings of fullness as markers of satiety, = (3) objective measures of energy intake following food intake, and (4) objective measures of body weight as a marker of weight loss.

The secondary outcomes included were (1) subjective ratings of preoccupation with food as a marker of appetite regulation, (2) objective measures of BMI, and (3) objective measures of WC as secondary markers of weight loss.

### 2.5. Quality Assessment

The quality of each included study was assessed using the Cochrane risk of bias tool, RoB 2 [24]. This quality tool assesses the risk of bias in 5 domains; the risk of bias classification was (a) low risk of bias; (b) some concerns; (c) high risk of bias. Two authors evaluated the risk of bias of each RCT (C.J.-t.H. and A.P.), and any disagreement between these authors regarding the risk of bias was resolved through discussion with the other authors.

## 3. Results

A total of 2829 articles were identified through the databases, 2330 were excluded due to duplication, 499 articles were screened, and 482 were excluded for not meeting the inclusion criteria. The remaining 17 articles were full-text-assessed for eligibility, and 10 were excluded for the following reasons: different study outcomes (n = 5), no results posted (n = 2), study design not accepted according to the eligibility criteria (n = 1), and not available online (n = 1). Finally, eight articles, describing nine RCTs (as one article reported two RCTs), were included in this systematic review [25,26,27,28,29,30,31,32], as shown in Figure 1. In Appendix A information regarding the excluded studies is shown.

### 3.1. The Characteristics of the Included Studies

Out of the nine RCTs, eight were RCT crossovers with no double blinding [25,26,27,28,29,31,32], and one study was a parallel, double-blind, controlled study [30].

In all the included studies, the study population included both sexes, and the subjects’ age was 18–50 years old. The sample size of the studies ranged from 33 to 201, and the duration of the intervention ranged from three days to eight weeks.

The studies were carried out in Europe [26,32] and America [25,27,28,29,30,31]. Out of the nine RCTs, seven evaluated the effect of chewing gum on appetite regulation [25,26,27,28,29,31,32], six RCTs evaluated the effect of chewing gum on satiety [25,26,27,28,29,32], seven RCTs evaluated the effect of chewing gum on energy intake [25,26,27,28,29,31], and one RCT evaluated the effect of chewing gum on weight loss [30], as shown in Table 2.

The characteristics and duration of the RCTs are very heterogeneous. The common traits among studies are that almost all, except one that was a parallel-controlled trial [30], were crossover studies, most described specific test days when volunteers attended the laboratory or medical center to complete the intervention, and participants received either a fixed meal before chewing gum or a snack 3 h after lunch.

The mastication of chewing gum oscillated between 10 and 20 min, from one to eight times a day, depending on the study protocol, and, in the no gum condition (control group), participants rested or stayed seated for the same amount of time. One RCT did not specify duration of chewing gum mastication [31].

In Appendix A, detailed information regarding the RCTs is shown.

The results of the RCTs are described as follows: Section 3.2. outlines chewing gum and appetite regulation (Section 3.2.1. relates to hunger; Section 3.2.2. describes desire to eat; Section 3.2.3. describes preoccupation with food); Section 3.3. relates to chewing gum and satiety; Section 3.4. relates to chewing gum and energy intake; Section 3.5. outlines chewing gum and weight loss.

### 3.2. Chewing Gum and Appetite Regulation

Appetite regulation was measured using hunger, desire to eat, desire to eat a sweet snack, desire to eat a salty snack, desire to eat a fatty snack, and preoccupation with food.

Out of the nine RCTs, seven studies evaluated the effect of chewing gum on hunger [25,26,27,28,29,31,32], five on the desire to eat [26,27,28,29,32], four on the desire to eat a sweet snack [26,27,28,32], four on the desire to eat a salty snack [26,27,28,32], and one on the desire to eat a fatty snack and preoccupation with food [27].

#### 3.2.1. Hunger

Of the seven RCTs, five RCTs reported that chewing gum had a significant suppressing effect on hunger compared to the control group [25,26,29,31,32], one RCT showed no statistical significance reduction [28], and one RCT described a stronger hunger sensation after gum chewing compared to the no gum chewing group [27].

In the study of Hetherington et al., participants ate a fixed lunch on arrival at the laboratory and chewed gum for 15 min at 1 h, 2 h, and 3 h after lunch, whereas under the no-gum conditions, participants rested for 15 min. After they chewed their last chewing gum, participants rated their hunger and had access to ad libitum snacks. The overall mean hunger ratings were significantly lower after gum chewing compared to after no gum chewing (hunger comparing gum condition to no gum: F (1,59) = 5.313, *p* = 0.025). The significant interaction between condition and time indicated that hunger ratings increased to a lesser extent after gum chewing compared to after not chewing gum (F (3,177) = 2.872, *p* = <0.005) [26].

In another study conducted by Hetherington et al., participants ate a fixed lunch on arrival at the laboratory and chewed gum for 15 min at 1 h, 2 h, 3 h after lunch and before snack intake; in the no gum condition, participants rested for the same time; hunger was rated after every gum chewing. Overall, hunger was observed to be lower in the gum condition compared to the no gum condition (hunger comparing gum condition to no gum: F (1,59) = 28.8; *p* < 0.001). The significant interaction between condition and time indicated that hunger ratings increased to a lesser extent after lunch to before the snack after gum chewing compared to after not chewing gum (F (3,177) = 10.09, *p* = < 0.001) [32].

In Swoboda et al., participants chewed gum for 10 min upon arrival to the laboratory and afterward played a reinforcement game to earn points to be exchanged for lower- or high-density snacks. Before eating the earned food, participants rated their hunger, and then they could eat as much or as less of the food that they earned. Hunger was significantly lower in the gum conditions, with no difference between fruit or mint gum, compared to the no gum conditions (hunger after chewing mint gum: F (3,129) = 5.9; *p* = 0.001; hunger after chewing fruit gum: F (3,129) = 5.5; *p* = 0.001; comparison of hunger between mint and fruit gum: F (3,129) = 0.49; *p* = 0.68) [31].

Additionally, in Park et al., participants ate a fixed lunch upon arrival at the laboratory and chewed gum for 15 min at 1 h, 2 h and 3 h after lunch. After they chewed their last chewing gum, participants rated their hunger and had access to ad libitum snacks. Hunger was significantly lower in the gum conditions compared to the no gum conditions (hunger comparison between groups: *p* = 0.006) [29].

Furthermore, in Bobillo et al., participants ate a fixed lunch upon arrival at the laboratory and chewed gum (either active gum or placebo gum) for at least 15 min every hour starting 45 min after breakfast and 1 h, 2 h, 3 h, and 4 h after lunch, for a total of eight pieces of gum. In the no gum condition, participants rested for at least 15 min every hour, instead of chewing gum. Hunger was rated before and after lunch every 30 min up to 4 h after lunch. A significant decrease in hunger was observed in the active and placebo gum conditions compared to the no gum condition, without differences between active and placebo gum, pre- and post-lunch (pre-meal ± SE mean difference between active gum versus no gum: −7.89 ± 2.96; *p* = 0.01; pre-meal mean ± SE difference between placebo gum versus no gum: −10.55 ± 2.96; *p* = 0.004; pre-meal mean ± SE difference between active gum and placebo gum: *p* = 0.41; post-lunch mean ± SE difference active gum versus no gum: −5.32 ± 2.25; *p* = 0.02; post-lunch mean ± SE difference placebo gum versus no gum: −5.83 ± 1.95; *p* = 0.004; post-lunch mean ± SE difference between active gum and placebo gum: *p* > 0.05) [25].

In Melanson et al., participants underwent a 45 min measurement of resting metabolic rate (RMR) using ventilated hood indirect calorimetry. Ten minutes into the RMR measurement, in the gum condition, participants chewed gum for 20 min; in the no gum condition, the passing and collection of gum were simulated. After the 45 min of RMR, participants received a standardized breakfast. The hood of the indirect calorimeter remained in place throughout the 3 h postprandial measurement, except for brief time periods when subjects rated their appetite as well as delivered and collected gum, or during which these processes were simulated in the no gum condition. In the gum condition, volunteers chewed gum two additional times for 20 min each time (60–80 min and 150–170 min after eating), for a total 60 min of chewing time during the morning of testing. At the end of the 3 h postprandial measurement period, subjects completed a final rating of appetite and were then offered an ad libitum pasta lunch with water. A significant reduction in hunger was observed after the first chewing period (hunger before first chewing period comparison between groups: t = 1.37; *p* = 0.18; hunger after first chewing period comparison between groups: t = 2.66; *p* = 0.01), but no significant changes were observed at 90 min and 180 min after eating when comparing the gum condition to the no gum condition (hunger 90 min postprandial comparison between groups: t = 0.05; *p* = 0.96; hunger 180 min postprandial comparison between groups: t = 0.78; *p* = 0.44) [28].

In Julis et al. on arrival at the laboratory, participants either chewed gum for 20 min after a 2 h fixed-time meal or after consuming a calorie-containing food or drink after lunch. Ratings of hunger were completed on arrival at the laboratory, after lunch, every 30 min after leaving the laboratory until the next eating occasion, after gum chewing, and after the intake of food and caloric beverages. The mean post-lunch hunger was higher in the gum condition compared to the no gum condition (mean ± SE post-lunch hunger for fixed-time gum treatment: 27 ± 2, *p* > 0.05; mean ± SE post-lunch hunger with pre-meal gum treatment: 29 ± 1, *p* > 0.05; mean ± SE post-lunch hunger, no gum 25 ± 2, *p* > 0.05), and hunger significantly increased after chewing gum (mean ± SE hunger fixed-time treatment before gum: 31 ± 2, *p* < 0.05; after gum: 32 ± 3, *p* < 0.05; mean ± SE hunger pre-meal before gum: 45 ± 3, *p* < 0.05; after gum: 48 ± 3, *p* < 0.05) [27].

#### 3.2.2. Desire to Eat

The desire to eat information included the desire to eat a sweet snack or a salty snack or a fatty snack.

Three out of the five RCTs that evaluated the desire to eat and chewing gum described a significant reduction in the desire to eat after chewing gum compared to not chewing gum [27,29,32]. One RCT described a positive correlation between the desire to eat, energy intake, and the gum condition [26], and one RCT described no effect on the desire to eat [28].

Hetherington et al. observed a significant decrease in the desire to eat after gum chewing compared to not chewing gum (desire to eat comparing gum condition to no gum condition (F (1,59) = 21.3, *p* < 0.001). A significant interaction between the condition and time indicated that the desire-to-eat ratings increased to a lesser extent after gum chewing compared to after not chewing gum (F (3,177) = 7.259, *p* < 0.001) [32].

Furthermore, Julis et al. described a reduction of desire to eat was seen in the fixed time gum treatment after the gum, however in the pre-meal treatment a significant increase in desire to eat after chewing gum was observed (mean ± SE desire to eat fixed time gum treatment before gum chewing: 31 ± 3 mm, *p* < 0.05; after gum chewing: 27 ± 3 mm, *p* < 0.05; mean ± SE desire to eat pre-meal gum treatment before gum chewing: 43 ± 3 mm, *p* < 0.05; after gum chewing: 46 ± 3 mm, *p* < 0.05) [27].

Additionally, Park et al. described a significant decrease in desire to eat was observed after chewing gum compared to no gum (desire to eat comparison in between groups: *p* = 0.002) [29].

Three out of the four RCTs that evaluated the desire to eat a sweet snack described a significant reduction after gum chewing compared to no gum chewing [26,27,32] and one described no effect on the desire to eat a sweet snack [28]. Hetherington et al. described that there was a significant reduction in desire to eat a sweet snack after gum chewing compared to no gum chewing (desire to eat a sweet snack comparing gum condition to no gum condition: F (3,177) = 4.530; *p* = 0.004) [26].

In another study conducted by Hetherington et al., a reduction of the desire to eat something sweet was observed after gum chewing compared to no gum (desire to eat something sweet in gum condition compared to no gum condition: F (1,59) = 22.5; *p* < 0.001) [32]. A significant interaction between condition and time indicated that ratings on the desire to eat a sweet snack increased to a lesser extent after gum chewing compared to no gum chewing: (F (3,177) = 2.7, *p* = 0.048).

Julis et al. observed a non-significant reduction in the desire to eat something sweet in the fixed-time gum treatment and pre-meal gum treatment comparing before and after gum chewing (mean ± SE desire to eat something sweet, fixed-time gum treatment before gum: 22 ± 3, *p* < 0.05; after gum: 21 ± 3, *p* > 0.05; mean ± SE desire to eat something sweet, pre-meal gum treatment before gum: 31 ± 3, *p* < 0.05; after gum: 29 ± 3, *p* > 0.05) [27].

Four RCTs evaluated the effect of chewing gum on the desire to eat a salty snack [26,27,28,32]; only one showed a significant reduction in the desire to eat a salty snack after chewing gum chewing compared to no gum chewing (desire to eat something salty in gum condition compared to no gum condition: F (1,59) = 20.6; *p* < 0.001) [32]. Additionally, a significant interaction between condition and time indicated that the ratings of the desire to eat a salty snack increased to a lesser extent after gum chewing compared to no gum chewing (F (3,177) = 2.9, *p* = 0.036) [32]. 

Only one RCT evaluated the effect of chewing gum on the desire to eat a fatty snack, and no effect was found [27].

#### 3.2.3. Preoccupation with Food

Only one RCT evaluated the effect of chewing gum on the preoccupation with food, and the results showed a significantly lower preoccupation with food after gum chewing in the fixed-time gum treatment and pre-meal gum treatment (mean ± SE preoccupation with food, fixed-time gum treatment, before gum: 25 ± 3, *p* > 0.05; after gum: 23 ± 3, *p* < 0.05; mean ± SE preoccupation with food, pre-meal gum treatment, before gum: 32 ± 3, *p* > 0.05; after gum: 36 ± 4, *p* < 0.05) [27].

### 3.3. Chewing Gum and Satiety

Six RCTs evaluated the effect of chewing gum on satiety measured as fullness [25,26,27,28,29,32]. Three studies found a significant increase in fullness in the chewing gum condition compared to the control group [25,26,32], two RCTs found a non-significant increase in fullness compared to no gum [28,29], and one RCT found a significant decrease in fullness after gum chewing [27].

Hetherington et al. reported a significant increase in fullness in the gum condition compared to the no gum condition (fullness comparing gum to no gum condition: F (1,59) = 4.545, *p* = 0.04) [26].

In another study conducted by Hetherington et al., fullness was significantly higher after gum chewing compared to no gum chewing (fullness in gum condition compared to no gum condition: F (1,59) = 5.18; *p* = 0.026). Furthermore, a significant interaction between condition and time indicated that the ratings of fullness decreased to a lesser extent after gum chewing compared to no gum chewing: F (3,177) = 3.96, *p* = 0.009 [32].

Lastly, Bobillo et al. described a significant increase in fullness after chewing either active and placebo gum when compared to no gum chewing before a meal (mean ± SE difference between active gum versus no gum: 7.89 ± 3.22 mm; *p* = 0.018; mean ± SE difference between placebo gum versus no gum: 5.80 ± 2.98 mm; *p* = 0.057) and after lunch (mean ± SE difference active gum versus no gum: 5.54 ± 2.50 mm; *p* = 0.03; mean ± SE difference placebo gum versus no gum: 6.12 ± 2.23 mm; *p* = 0.008; active gum versus placebo gum: *p* > 0.05) [25].

However, Julis et al. found a significantly lower sensation of fullness in the gum condition between before and after gum chewing (mean ± SE fullness fixed-time gum treatment before gum: 51 ± 3 mm, *p* < 0.05; after gum 47 ± 4 mm, *p* < 0.05; mean ± SE fullness pre-meal gum treatment before gum: 37 ± 3 mm, *p* < 0.05; after gum: 34 ± 2 mm, *p* < 0.05), but no information on the control group was given [27].

### 3.4. Chewing Gum and Energy Intake

Seven out of the nine included RCTs evaluated the effect of chewing gum on energy intake [25,26,27,28,29,31]. Only two RCTs reached significance in the reduction in energy intake in the gum condition compared to the control group [26,32].

Hetherington et al. described a reduction in energy intake in the gum condition compared to the no gum condition (energy intake comparing gum to no gum condition: F (1,58) = 4.344, *p* = 0.04) [26].

Similar findings were described by Melanson et al., where energy intake was lower in the gum condition compared to the no gum condition (energy intake comparison between groups: t = 3.130; *p* = 0.004) [28].

Four studies did not find any significant effect on energy intake. Swoboda et al. described a significant reduction in the energy intake of healthy food but not for total daily energy intake in the gum condition when compared to the no gum condition (interaction between gum chewing and intake of healthy food: F (2,86) = 5.8; *p* = 0.004) [31].

Furthermore, in a second RCT conducted by Swoboda et al., a significant reduction in energy intake per meal was observed (chewing gum and energy intake per meal: F (2,106) = 10.3; *p* < 0.0001), but no significance changes in the total daily energy intake were observed in the gum condition compared to the no gum condition [31].

Additionally, Park et al. described a significant reduction in carbohydrate intake in the gum condition compared to no gum but a non-significant reduction in energy intake from snacks (carbohydrate intake; total snack intake comparison between groups: *p* = 0.08) [29].

Lastly, Bobillo et al. described a significant reduction in the total energy intake from snacks for the active gum when compared to no gum, but no effects were found for the placebo gum when compared to no gum (total energy intake from snacks comparing active gum and no gum: *p* < 0.001; total energy intake from snacks comparing placebo gum and no gum: *p* > 0.05) [25].

### 3.5. Chewing Gum and Weight Loss

Only Shikany et al. assessed the effect of chewing gum on weight loss and BMI, finding no effects of the chewing of gum on weight loss or BMI compared to no gum chewing. A significant intertreatment reduction in WC was observed in the gum condition, although no significant reduction was observed when compared to the no gum condition (mean ± SD change in WC in the gum condition: −1.7 ± 5.7 cm; *p* < 0.05; mean ± SD change in WC in the no gum condition: −0.7 ± 5.5 cm; *p* > 0.05; comparison between groups: *p* = 0.27) [30].

### 3.6. Quality of the RCTs Included in This Systematic Review

According to the Cochrane risk of bias tool RoB2 [24], of the nine RCTs included, one was classified as having a low risk of bias [30] and eight as having some concerns in domain 1 (randomization process) [25,26,27,28,29,31,32] (Figure 2).

## 4. Discussion

The present systematic review showed that chewing gum has an influence on appetite regulation by reducing hunger, the desire to eat, and the desire to eat a sweet snack, but the effect on the desire to eat a salty and a fatty snack, preoccupation with food, satiety measured as fullness, energy intake, and weight loss is not conclusive.

Our results suggest that the mastication effect of chewing gum has an impact on appetite regulation and that there might be a suppressor effect on appetite sensations, specifically on hunger. Similar findings were observed in a systematic review and meta-analysis [16], stating that prolonged chewing, regardless of whether the mastication of food or sham feeding with chewing gum, reduces self-reported hunger. Additionally, in a meta-analysis, Krop et al. concluded that prolonged orosensory exposure to food reduced subjects’ appetite and increased chewing reduced food intake [17]. Furthermore, in a study conducted by Ikeda et al., chewing stimulation reduced subjective appetite, suggesting that chewing, even without ingestion, may affect reward circuits and reduce subjective appetite ratings that reflect cravings [20]. The consistency across these trials highlights the potential utility of gum chewing as a non-pharmacological intervention to control hunger.

Reducing the consumption of sweet or salty snacks is a key strategy for managing obesity. These foods are often high in energy, low in nutrients, and easily accessible, which contribute to excessive energy consumption and, consequently, weight gain [33,34]. These findings suggest that chewing gum could serve as an effective measure to curb general cravings and manage overall food intake. In a systematic review conducted by Cooke et al., discretionary snack consumption was associated with energy intake, suggesting that the increased consumption of snacks might contribute to increased energy intake; however, there was a lack of consistent associations with increased weight and BMI [35].

Concerning satiety, measured as fullness, three out of six RCTs showed a significant increase in fullness after chewing gum when compared to no gum chewing [25,26,32]. However, one RCT found a significant decrease in fullness between before and after chewing gum chewing [27]. These mixed results suggest that chewing gum might influence satiety signals; however, the non-significant results in these studies might be due to the variations in study design, participant characteristics, or sensitivity of the measures used to assess fullness. A study conducted by Komai et al. described tendencies toward increases in satiety and fullness after 20 min of gum mastication [36]. Similar results were observed in another study where chewing stimulation, via chewing gum, was compared to actual feeding to reduce attentional bias toward food and concluded that actual feeding significantly increased fullness. Chewing gum showed the same results but to a lesser extent [20]. However, a systematic review and meta-analysis by Miquel-Kergoat et al. showed that only a minority of studies demonstrated significant increases in fullness and satiety after chewing without considering if it was food or sham feeding [16].

Increasing satiety is a crucial factor in obesity management, as it can help in controlling food intake and contribute to weight loss and the maintenance of a healthy weight [37]. Therefore, more high-quality interventions are needed to elucidate the effectiveness of chewing gum on increasing satiety.

Regarding appetite, increased masticatory cycles, either by thorough the mastication of food or sham feeding with chewing gum, have been described to reduce postprandial appetite and influence appetite hormones such as GLP-1, CCK, and ghrelin [14,38,39]. In a study conducted with 12 healthy male volunteers, a decrease in GLP-1 concentration was associated with an increase in satiety after 30 min of chewing gum [38]. Various studies associated an increase in CCK and a reduction of ghrelin after an increased number of chewing cycles before swallowing food [14,39]. These results show that thorough mastication may increase satiety.

These results and the results obtained in this systematic review show that chewing gum might influence appetite regulation and, consequently, may increase satiety and therefore reduce energy intake. These findings suggest that chewing gum could enhance feelings of fullness in various experimental setups, particularly when consumed in regular intervals after meals. Taking into account that current antiobesity drugs target satiety signaling in the brain, often do not produce the expected effect, can cause chronic disorders and side effects, and do not have a sufficient level of safety [40], the use of chewing gum to increase satiety signals would be a novel tool.

Concerning energy intake, the mixed findings across these studies suggest that while chewing gum may help reduce energy intake under certain conditions, its overall impact on daily energy consumption is inconsistent [26,28,31]. Factors such as the timing of gum chewing, types of meals, and individual differences likely play critical roles in these outcomes [41]. Furthermore, only one RCT was a long-term study; all other RCTs were acute studies. This suggests that the results might have been influenced, and it would be interesting to carry out more long-term studies that evaluate energy intake in order to see changes over time.

The evidence collected and the current bibliography suggest that chewing gum could be a novel tool to reduce hunger and desire to eat, whereas its effect on body weight has not been proven. These findings support our hypothesis stating that chewing gum affects appetite regulation, specifically hunger and desire to eat, but the effects on satiety and energy intake are inconclusive. Taking into account that there are already drugs on the market that are used for weight loss, our results suggest that chewing would have the same influence as these drugs, with lesser side effects. However, more studies with a longer duration are needed to evaluate the chronic effect of chewing gum to establish a frequency of chewing gum chewing that does not cause side effects such as abdominal distention and to analyze whether the decreases in hunger and desire to eat translate into a decrease in energy intake and ultimately decreases in weight, BMI, and WC.

To the best of our current knowledge, this is the first systematic review that has explicitly assessed the influence of chewing gum on satiety, appetite regulation, and energy intake in RCTs.

One of the key limitations of this systematic review is the heterogenicity of the studies. The differences in the intervention design, the evaluation of the outcomes, and the duration of the intervention make it very difficult to compare these studies and establish conclusions. There is a need to carry out intervention studies with the same type of blinding, duration of intervention, and outcomes to be studied to be able to draw conclusions and define the role of chewing gum on appetite regulation and satiety. The purpose of these studies should be to conduct randomized controlled studies that evaluate the effect of chewing gum on satiety and appetite, as well as chewing it for a certain time and a certain period of time, compared to not chewing gum. Comparisons between chewing gum and actual food should be avoided, since the hormonal responses differ [6]. These studies are essential to determine whether chewing gum is an effective tool for obesity and to be able to use it in clinical practice. Lastly, most of the articles included in this review lacked statistical data, such as mean differences, standard deviation, and standard error or confidence intervals for each intervention, as well as their *p*-values. Consequently, a meta-analysis, which would have provided more conclusive results, as well as a forest plot, which would have provided a clearer presentation of the results, could not be performed. Finally, we also point out that the data were extracted from the articles by only one researcher, and although a second researcher was consulted in case of doubt, the fact of not performing the entire extraction independently by at least two researchers could have led to biases.

Further research on the effect of chewing gum on populations with overweight, obesity, or abdominal obesity is needed to establish if it is a good co-adjuvant tool for weight loss and, consequently, for obesity management.

Future research should aim to elucidate the mechanisms, optimal conditions, and long-term effects of gum chewing on appetite and weight management.

## 5. Conclusions

The results of this systematic review confirm that chewing gum influences appetite regulation parameters, particularly leading to a clear decrease in the feeling of hunger, desire to eat, and desire to eat a sweet snack. While the findings regarding chewing gum are promising, further research to reduce the heterogenicity between RCTs is needed to elucidate the mechanisms, optimal conditions, and long-term effects of gum chewing on appetite and weight management.

## Figures and Tables

**Figure 1 nutrients-17-00435-f001:**
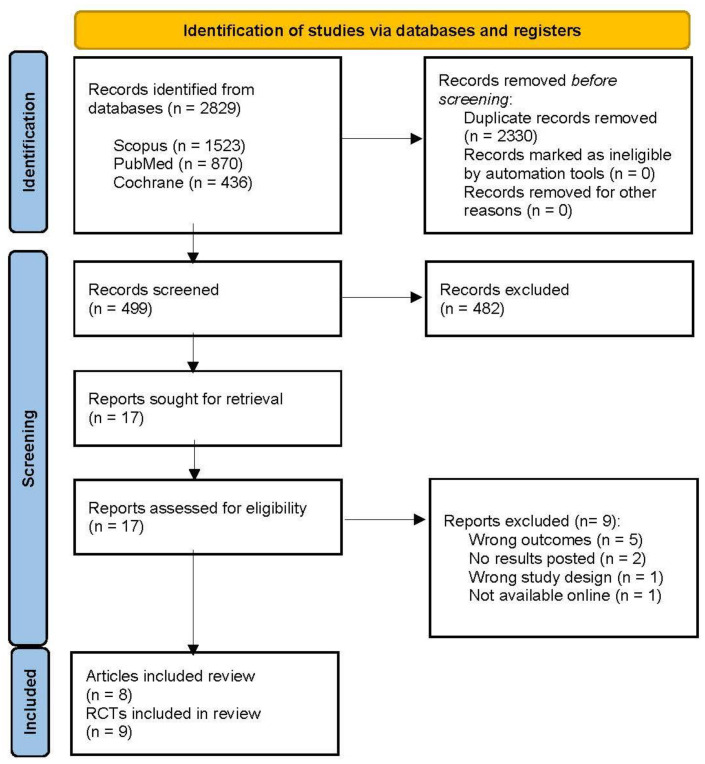
PRISMA flow diagram of the study selection procedure.

**Figure 2 nutrients-17-00435-f002:**
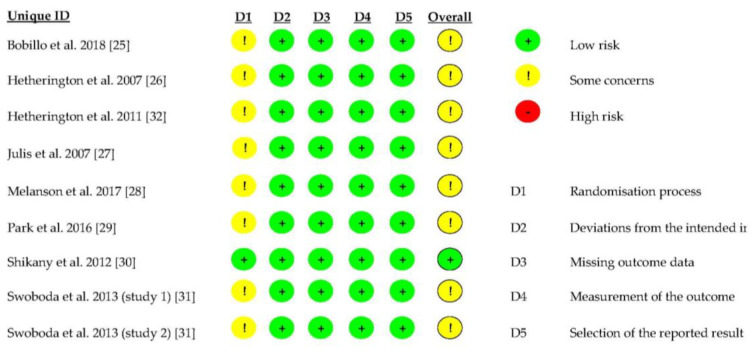
Quality of the RCTs included in this systematic review according to the Cochrane risk of bias tool RoB2.

**Table 1 nutrients-17-00435-t001:** PICOS criteria for eligibility of RCTs.

Criteria	Inclusion	Exclusion
Population	- Adults over 18 years old - All sexes and races - Healthy adults or with some comorbidities	- Children and adolescents (<18 years old) - Old adults (>65 years old) - Individuals with a specific health condition
Intervention	- Interventions that analyzed the effect of chewing gum on satiety, appetite regulation, energy intake, and/or weight loss - Any low-calorie chewing gum type - Sustained, postprandial, or short-term interventions	- Studies not involving chewing gum interventions - Studies assessing the effect of chewing gum on other outcomes such as concentration and alertness - Studies involving medicated chewing gum interventions
Comparison	- No use of chewing gum or avoiding consumption	- Consumption of low-calorie food - Sham feeding
Outcomes	- Changes in satiety and/or appetite regulation - Changes in energy intake or caloric intake - Weight loss or changes in body weight	- Articles that report no relevant data of the mentioned outcomes
Study type	- Randomized clinical trials (RCTs) involving controlled interventions - Parallel or crossover design	- Systematic reviews and/or meta-analyses - Case–control studies - Cohort studies - Protocols

RCTs: Randomized controlled trials.

**Table 2 nutrients-17-00435-t002:** Characteristics of the RCTs included in this systematic review addressing the effect of chewing gum on appetite regulation, satiety, energy intake, and weight loss in adults.

Author	Participants		Intervention	Results			
	N	BMI	Design	Effect on Appetite Regulation	Effect on Satiety	Effect on Energy Intake	Effect on Weight Loss
Hetherington and Boyland(2007) [26]	60	NOB or OW	Fixed lunch on arrival. Gum chewed 1 h, 2 h, and 3 h after lunch. Access to ad libitum snacks 3 h after lunch. Intake recorded at home by diet records; within subjects.	Hunger * and desire to eat sweet snacks * ↓ in the chewing gum condition. No effect on desire to eat salty snacks.	Fullness * ↑ in the chewing gum condition.	Energy intake * ↓ in the chewing gum condition.	NA
Julis and Mattes(2007) [27]	47	OW	Standard breakfast and lunch. Gum chewed at a fixed time or before meal for 20 min before food consumption. Intake recorded at home by diet records; within subjects.	Hunger ^α^ ↑ after gum chewing, but desire to eat ^α^ and desire to eat sweet snacks ↓ in the chewing gum condition. No effect on salty snacks, fatty snacks, or preoccupation with food.	Fullness ^α^ ↓ after gum chewing.	Energy intake ↓ in the chewing gum condition.	NA
Hetherington and Regan(2011) [32]	60	NOB or OB	Fixed lunch on arrival. Gum chewed 1 h, 2 h, and 3 h after lunch. Access to ad libitum snacks 3 h after lunch. Intake recorded at home by diet records; within subjects.	Hunger **, desire to eat **, desire to eat sweet snacks **, desire to eat salty snacks ** and desire to eat a snack ** ↓ in the chewing gum condition.	Fullness * ↑ in the chewing gum condition.	NA	NA
Shikany J et al. (2012) [30]	201	OB	Eight-week intervention with chewing gum and printed nutrition information. Participants chewed gum six times a day for a total of 90 min/day (20 min after breakfast, lunch, and dinner and chewed an additional 10 min mid-morning, mid-afternoon, and 1–2 h after dinner).	NA	NA	NA	No effect on weight loss.
Swoboda and Temple (study 1) (2013) [31]	44	OW	Gum chewed for 10 min upon arrival at the laboratory. Reinforcement game to earn points in order to earn food (lower or high energy density). Intake recorded at home with diet records; within subjects.	Hunger ^β^ was ↓ in the chewing gum condition.	NA	Energy intake of healthy food * ↓ in the chewing gum condition but no effect on total daily energy intake.	NA
Swoboda and Temple(study 2) (2013) [31]	54	OW	Gum chewed every single occasion before food for a week. Intake recorded at home using diet records; within subjects.	NA	NA	Energy intake per meal * ↓ in the chewing gum condition but no effect on total daily energy intake.	NA
Park et al.(2016) [29]	50	NOB and OB	Fixed lunch on arrival. Gum chewed 1 h, 2 h, and 3 h after lunch. Access to ad libitum snacks 3 h after lunch. Intake recorded at home using diet records; within subjects.	Hunger * and desire to eat * ↓ in the chewing gum condition.	Fullness ↑ in the chewing gum condition.	Energy intake from snacks ↓ in the chewing gum condition. Carbohydrate intake * was ↓ in the chewing gum condition.	NA
Melanson and Kresge(2017) [28]	33	NOB or OW	Fixed breakfast on arrival. Gum chewed for 20 min, 10 min into ventilated hood indirect calorimetry, and 3 h after breakfast. Access to ad libitum snacks 3 h after breakfast. Intake recorded at home using diet records; within subjects.	Hunger * ↓ in the chewing gum condition only after first chewing period. No effect on desire to eat, desire to eat something sweet, or desire to eat something salty.	Fullness ↑ in the chewing gum condition.	Energy intake * ↓ in the chewing gum condition.	NA
Bobillo et al.(2018) [25]	57	NOB or OW	Fixed lunch on arrival. Gum chewed 1 h, 2 h, 3 h, and 4 h after lunch. Access to ad libitum snacks 4 h after lunch. Intake recorded at home in diet records; within subjects.	Hunger * ↓ in the chewing gum condition.	Fullness * ↑ in the chewing gum condition.	Total energy intake from snacks * ↓ for active gum when compared to no gum. No effect of placebo gum.	NA

NOB = participants without obesity; OW = participants with overweight; OB = participants with obesity; NA = not addressed; ↑ = increase; ↓ = decrease; * *p* < 0.05 comparing gum to no gum chewing; ^α^
*p* < 0.05 comparing before and after gum chewing; ** *p* < 0.001 comparing gum to no gum chewing; ^β^
*p* < 0.001 comparing before and after gum chewing.

## Data Availability

The data presented in this study are available upon request from the corresponding author due to privacy reasons.

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
