# Peer review of "Effects of Chewing Gum on Satiety, Appetite Regulation, Energy Intake, and Weight Loss: A Systematic Review"

_nutrients, 2025, doi:10.3390/nu17030435_

Round 1
Reviewer 1 Report
Comments and Suggestions for Authors
Review of “Effects of chewing gum on satiety, appetite regulation, energy intake and weight loss: a systematic review”
This is an interesting paper with a novel investigation. The paper is well-written, the authors present their rationale for the paper satisfactorily, and the data support their conclusion. But some aspects of the methods and reporting in general could be improved, as I point out below:
- According to PRISMA, authors must present the full search strategy exactly as conducted in order to allow replication. The authors claim that the search strategy is in the supplementary material, but this reviewer was not able to see the supplementary material in the submission files.
Regarding the search protocol as a whole, I have identified some issues that decrease the quality of the paper, and the authors should address them:
- The authors only searched three databases, and they make no mention of searching in trial registries or gray literature. It is advisable by most organizations (Cochrane, JBI…) that such sources must be searched, even if it results in no additional paper to be included in the study.
- Authors used several “filters” in their search strategy, such as date filters (only studies from 2000…) and language filters (only studies in English). Furthermore, it seems like the authors used filter options built-in the databases (such as “human only” and “study type”). There are several issues with this approach, but basically, it is not advisable to use language and date limits without a clear rationale. Also, it is not advisable to use filters built into the databases (such as “human only” and those referring to “study type”), given that they are too specific. There are several validated search strategies for searching RCT (such as the “Cochrane RCT search strategy), which is superior to using built-in filters.
- authors should cite a reference endorsing the use of data extraction by only one researcher. Most organizations suggest that two independent researchers must extract sensible data.
- Cochrane suggests that full texts retrieved but not included in the final study must be present in a supplementary table with all information of the article.
Reporting:
- Why study reference n. 25 is considered a “prospective crossover-controlled study”? What is the difference between this classification and the other cross-over included? Aren't all cross-over studies prospective by nature?
- I firmly believe that Table 2 could be split into 2 different tables, one with the characteristics of the design (% of males and females, if it was parallel or cross-over, country, how the kcal of the meals was calculated, mean age and mean BMI…) and the column “design” should be further split. Then, another table with the “results” could be shown separately, and it could bring numbers of the results (mean decrease in appetite, etc for both groups etc).
The authors say that “In Supplemental Table 2, detailed information regarding the RCTs is shown.” But I was not able to find this table.
- The authors state that “All eight RCTs presented some concerns in domain 1 (randomization process).” However, Shinkay 2011 did not. Please adjust it.
- In the paragraph beginning in line 496, the authors should explain in more detail how future studies should proceed. Authors only say that “same type of blinding, the same type of intervention”, but they do not say what specific type future studies should investigate.
Author Response
Thank you very much for taking the time to review this manuscript. Please find the detailed responses below and the corresponding revisions/corrections highlighted in the re-submitted files.
Comment 1: According to PRISMA, authors must present the full search strategy exactly as conducted in order to allow replication. The authors claim that the search strategy is in the supplementary material, but this reviewer was not able to see the supplementary material in the submission files.
We attach again the supplementary material with the search strategy information. As detailed in Supplementary Table 1, we searched in PubMed, SCOPUS, and Cochrane Central Register of Controlled Trials. The following search strategy was applied:
#1 (“chewing gum” OR “chewing-gum)
#2 (“chewing” OR “mastication”)
#3. (“appetite regulation” OR “appetite” OR “satiety regulation” OR “satiety” OR “hunger”)
#4. (“body weight” OR “body mass index” OR “weight loss” OR “waist circumference” OR “obesity”)
#5. (“energy intake” OR “caloric intake”)
|
Search |
Strategy of search |
|
1 |
1 and 3 |
|
2 |
1 and 3 and 4 |
|
3 |
1 and 3 and 4 and 5 |
|
4 |
2 and 3 |
|
5 |
2 and 3 and 4 |
|
6 |
1 and 3 and 4 and 5 |
Comment 2: Regarding the search protocol as a whole, I have identified some issues that decrease the quality of the paper, and the authors should address them:
The authors only searched three databases, and they make no mention of searching in trial registries or gray literature. It is advisable by most organizations (Cochrane, JBI…) that such sources must be searched, even if it results in no additional paper to be included in the study.
When the search for eligible studies was carried out, one of the databases employed was Cochrane Central Register of Controlled Trials, which is considered as a database where registries can be found. During the search, three eligible registries were found. These trials were already taken into account in the screening of eligible studies. Two did not present results, and one has been considered as a duplicate since the RCT is included in the review. All this information has been introduced in the PRISMA flow diagram (Figure 1).
Grey literature is an exclusion criterion of this systematic review as it is defined as material and research produced by organizations outside of the traditional commercial or academic publishing and distribution channels. Common grey literature publication types include reports (annual, research, technical, project, etc.), working papers, government documents, white papers and evaluations. Thus, RCTs, needed for this systematic review, couldn’t be found in grey literature.
Comment 3: Authors used several “filters” in their search strategy, such as date filters (only studies from 2000…) and language filters (only studies in English). Furthermore, it seems like the authors used filter options built-in the databases (such as “human only” and “study type”). There are several issues with this approach, but basically, it is not advisable to use language and date limits without a clear rationale. Also, it is not advisable to use filters built into the databases (such as “human only” and those referring to “study type”), given that they are too specific. There are several validated search strategies for searching RCT (such as the “Cochrane RCT search strategy), which is superior to using built-in filters.
As detailed in comment 1 and Supplementary material, in the search strategy no built-in filters were applied, it was a mistake. However, to include studies must be developed in human and based on RCT type.
The following sentence has been removed from the manuscript in order to avoid confusion (pg 3, ln 106-108): “We limited the search strategy to the years of publication from 2000 to 2024, the English language, human studies, and publication type.”
Comment 4: authors should cite a reference endorsing the use of data extraction by only one researcher. Most organizations suggest that two independent researchers must extract sensible data.
The authors thank the reviewer for this appreciation and we have better detailed this information in the final version of the manuscript so that it does not lead to confusion.
Specifically, two independent researchers screened all titles and abstracts for inclusion, however only one did the data extraction form the included studies.
The following sentence was added to the manuscript in order to avoid confusion (pg 4, ln 136-138): “All titles and abstracts were screened for inclusion by two independent researchers (C.J.-t.H. and A.P.). After this first screening, full-text studies were read and assessed according to predefined inclusion criteria by one independent researchers (C.J.-t.H.).”
And the following sentence was removed (pg 4, ln 138-139): “One researcher performed the data extraction (C.J.-t.H.).”
Also, considering that Cochrane recommended that data extraction should be carried out by two independent researchers, we introduced a limitation in pg 16, ln 515-518 :“Finally, we can also point out that the extraction of data from the articles was done by only one researcher, and although a second researcher was consulted in case of doubt, the fact of not doing the entire extraction independently by at least two researchers could have led to biases.”
Comment 5: Cochrane suggests that full texts retrieved but not included in the final study must be present in a supplementary table with all information of the article.
As the reviewer suggested, we added a supplementary table with the summary characteristics of the excluded studies (Supplementary Table 2) following the Cochrane guidelines.
We attach the supplementary table as a separate document in the response to reviewer’s platform.
Comment 6: Reporting: Why study reference n. 25 is considered a “prospective crossover-controlled study”? What is the difference between this classification and the other cross-over included? Aren't all cross-over studies prospective by nature?
We used the terminology described in the study. We eliminate the word “prospective” to eliminate confusion and defined it as a crossover-controlled study. The following changes were introduced in the manuscript (pg 5, ln 177-179): “Out of the nine RCTs, eight were RCT, crossovers with no double-blind [25–29,31,32], one was a prospective crossover-controlled study and one study was a parallel, double-blind, controlled study [30].”
Comment 7: I firmly believe that Table 2 could be split into 2 different tables, one with the characteristics of the design (% of males and females, if it was parallel or cross-over, country, how the kcal of the meals was calculated, mean age and mean BMI…) and the column “design” should be further split. Then, another table with the “results” could be shown separately, and it could bring numbers of the results (mean decrease in appetite, etc for both groups etc).
The authors say that “In Supplemental Table 2, detailed information regarding the RCTs is shown.” But I was not able to find this table.
Following the instructions of the reviewers, we split the table 2 in two difference tables (one with characteristics of the design and one with study results) and attach them in the response of the reviewers as a separate document file.
However, the authors consider that they are too extensive and therefore the initial proposal was to include a summarized table in the manuscript (table 2) and a supplementary table with all detailed information regarding characteristics of the studies and results.
We newly attach Supplementary Table 3 for revision in the response of the reviewer as a separate document file.
Comment 8: The authors state that “All eight RCTs presented some concerns in domain 1 (randomization process).” However, Shinkay 2011 did not. Please adjust it.
In the present review, there are nine RCTs included from eight articles. Eight RCTs presented some concerns in domain 1. Only RCT Shikany 2011 did not.
The manuscript has been modified as follow (pg 14, ln 411-412):
“According to the Cochrane risk of bias tool RoB2 [24], of the nine RCTs included, one was classified as having a low risk of bias [30] and eight as having some concerns pre-sented in domain 1 (randomization process) [25–29,31,32] (Figure 2).”
Comment 9: In the paragraph beginning in line 496, the authors should explain in more detail how future studies should proceed. Authors only say that “same type of blinding, the same type of intervention”, but they do not say what specific type future studies should investigate.
More information specifying how these future studies should investigate the relationship between chewing gum and appetite regulation and satiety has been included.
The manuscript has been modified, as follows (pg 16, ln 504-508):
“The purpose of these studies should be to conduct randomized controlled studies that evaluate the effect of chewing gum on satiety and appetite, chewing it for a certain time and a certain period of time, compared to not chewing gum by. Comparisons between chewing gum and actual food should be avoided, since the hormonal response will be different [6]”.

Reviewer 2 Report
Comments and Suggestions for Authors
This is a very interesting and worth publishing systematic review concerning the effect of chewing gum on satiety, appetite regulation, energy intake and weight loss. Authors followed appropriate protocol concerning such type of studies.
What I missed in this paper was short description about excluded reports (Fig. 1). What decided about wrong outcomes, wrong study design or wrong intervention? Short description would be helpful to understand why those studies were excluded and dispel the reader's doubts that the review is not biased.
Author Response
Thank you very much for taking the time to review this manuscript. Please find the detailed responses below and the corresponding revisions/corrections highlighted in the re-submitted files
Comment 1: What I missed in this paper was short description about excluded reports (Fig. 1). What decided about wrong outcomes, wrong study design or wrong intervention? Short description would be helpful to understand why those studies were excluded and dispel the reader's doubts that the review is not biased.
Following a reviewer suggestion, a Supplementary Table (S2) has been elaborated with all the excluded studies following the Cochrane guidelines.
The terminology used to describe the reasons of exclusion follows Cochrane guidelines. We add the description of this terms in the Footnote of the Supplementary Table 2, find this file attached to the response to reviewers platform.
After reviewing the data, we realized that one study excluded for not having results is actually a duplicate. The corresponding modifications have been made both in the manuscript, PRISMA flow diagram (Figure 1) and supplementary table.
